# Two Fascinating Polysaccharides: Chitosan and Starch. Some Prominent Characterizations for Applying as Eco-Friendly Food Packaging and Pollutant Remover in Aqueous Medium. Progress in Recent Years: A Review

**DOI:** 10.3390/polym13111737

**Published:** 2021-05-26

**Authors:** Nancy Alvarado, Romina L. Abarca, Cristian Linares-Flores

**Affiliations:** 1Instituto de Ciencias Químicas Aplicadas, Facultad de Ingeniería, Universidad Autónoma de Chile, El Llano Subercaseaux 2801, San Miguel 8900000, Chile; 2Departamento de Ciencias Animales, Facultad de Agronomía e Ingeniería Forestal, Pontificia Universidad Católica de Chile, Macul, Santiago 7820436, Chile; romina.abarca@uc.cl; 3Grupo de Investigación en Energía y Procesos Sustentables, Instituto de Ciencias Químicas Aplicadas, Facultad de Ingeniería, Universidad Autónoma de Chile, El Llano Subercaseaux 2801, San Miguel 8900000, Chile; cristian.linares@uautonoma.cl

**Keywords:** biopolymers, eco-friendly, food packaging, bio-adsorbent, physical characterization, waste

## Abstract

The call to use biodegradable, eco-friendly materials is urgent. The use of biopolymers as a replacement for the classic petroleum-based materials is increasing. Chitosan and starch have been widely studied with this purpose: to be part of this replacement. The importance of proper physical characterization of these biopolymers is essential for the intended application. This review focuses on characterizations of chitosan and starch, approximately from 2017 to date, in one of their most-used applications: food packaging for chitosan and as an adsorbent agent of pollutants in aqueous medium for starch.

## 1. Introduction

The use of petroleum-based materials has been very effective and has improved our quality of life. However, the cost of our comfort is being paid by the environment due to the slow degradation of these materials, causing a huge accumulation of waste. It is urgent to change these materials of slow degradation and look for eco-friendly ones. In this research, chitosan (CS) and starch, two polysaccharides, have been extensively studied with the following purpose: to reduce the wastes from non-biodegradable materials [1,2].

After cellulose, chitosan is the most abundant biopolymer on our planet. This polysaccharide comes from the deacetylation of chitin, which is mainly present in the shell of crustaceans, insects, and marine invertebrates [1,3]. Worldwide, the production of cultured crustaceans reaches 11%, which corresponds to 9.4 million tons (Figure 1) [4]. In this sense, this industry originates large amounts of wastes containing chitin, therefore, representing an important source of CS from waste that could be used. Meanwhile, starch is a biopolymer that can be obtained from plants and plant wastes such as tuber wastes. The worldwide production of products containing starch is huge, generating around 2000 million tons (Figure 2) [5]; thus, the wastes produced when obtaining starch can be a formidable source of this biopolymer. Its cheap and abundant qualities make it an attractive material to both academia and industry.

Regarding the chemical structure of chitosan and starch, due to their nature as polysaccharides, both present –OH groups, and CS has –NH_2_ groups. The presence of these chemical groups is good since they allow chemical modifications. This characteristic gives versatility to these biopolymers as it could improve several physical characteristics and thereby extend the range of applications of these biopolymers. For instance, CS—modified through the introduction of imine and pyridine groups—was used as bioadsorbent of Cr(IV) and Cu(II) [6]. Meanwhile, when starch is treated with kaolin at high temperature, a starch with improved biodegradability properties is obtained [2]. Some of their applications are encapsulating agent [7,8,9], bio-adsorbent [10,11,12,13,14,15,16,17,18], foams [19,20,21,22,23,24], food packaging [25,26,27,28,29,30,31,32,33], and medical applications [34,35,36,37,38,39,40,41,42] (Figure 3).

Currently, it is well known that chemical modifications provide different, new, and improved materials. The blending between polymers, biopolymers, or composites is a cheap alternative that allows obtaining the desired material through the interactions. In this way, chitosan and starch have also been involved in several studies using this technique. These studies include the combination of biopolymers [43,44,45,46,47,48,49,50,51,52] and/or the incorporation of fillers, such as nanoparticles, fibers, and layers [47,48,49,50,51,52,53,54,55,56,57,58,59,60,61,62,63,64,65,66,67,68,69,70,71].

The antimicrobial capacity of CS, along with being biocompatible and non-toxic, provides unique features to this biopolymer which has been studied in depth with the goal of being applied in various fields such as pharmaceutical [72], medicine [41], food industry [73], etc. This review approaches some characterizations that describe and support classic methods concerning the use of chitosan in the food packaging industry and starch as pollutants remover. With this, it is expected to achieve a better understanding of the physical properties of these biopolymers in these specific fields.

## 2. Chitosan (CS)

The focus on biopolymers such as CS has increased due to the shortage of the fossil fuel reserve and the environmental impact induced by the accumulation of non-biodegradable plastic-based traditional packaging materials. This serious problem affects not only marine life but also humans, as microplastics can enter the food chain and eventually harm health [74]. It has been predicted that the global turnover of bioplastic manufacturing will increase 2.6 times from 2017 to 2022, reaching US $43.87 [75]. Currently, different biopolymers are obtained from food waste, which contributes to reducing “waste and at the same time diversifying food packaging materials. Thereby, CS is an alternative to synthetic polymers and a raw material for new materials” [76]. It is a natural material that has undergone widespread research in recent years. Most biopolymers, such as CS, are inherently weak regarding their physical, mechanical, and thermal attributes when compared to widely used petro-based polymers (i.e., polyethylene, polyvinyl alcohol, polystyrene) [77]. In the specific case of chitosan-based compounds, they become relevant because of their applications in food packaging and because they can be used in different ways [78,79,80].

The use of biopolymers on a large scale is still limited, “due to its high cost and lower performance based on poor barrier against water vapor; lower thermal, optical and mechanical properties when compared to traditional petroleum-based plastics” [81]. The versatile structural and magnificent characteristic of CS, as well as chemical modifications and nanobiotechnology creativity, permit the development of a series of chitosan-based nanomaterials and composites with interesting properties [82]. Below are a series of analyses used to characterize chitosan-based films, including their mechanical, physical, and thermal properties, among others.

### 2.1. Mechanical Properties of CS

Investigating the mechanical properties in food packaging is very important since, by knowing them, their adequate application is possible. The requirements concerning the mechanical properties of packaging are good tensile strength, flexibility, necessary elongation, and resistance to drilling, to generate a physical barrier and providing a longer shelf life to the food [83]. To verify that a material possesses good characteristics for use in food packaging, one or more of the reference parameters listed below can be used: maximum stress (tensile strength, TS) (MPa), Young’s modulus of elasticity (EM) (MPa) from the stress-strain curves, and film extensibility (elongation at break, %EAB) from the force–distance curves are determined using Equations (1)–(3), respectively.
(1)$$TS\left(MPa\right)=F_{max}/A_{0}$$
(2)$$EAB\;\left(\%\right)\;=\;\left[\Delta l/l_{0}\right]\times 100$$
(3)$$EM=L_{0\;}\left(F_{2}-F_{1}\right)/A_{0\;}\left(\delta_{2}-\delta_{1}\right)$$
where $F_{max}$ is the maximum load for breaking the film (N), Ao is the initial specimen cross-sectional area (T × W mm^2^), and Δ*l* is the change in length from the original length (*l_0_*) of the specimen between the grips. The force (*F*_1_, *F*_2_) and displacement (*δ*_1_, *δ*_2_) values were obtained using the linear portion of a plot of displacement against force [84].

CS film with various concentrations of green-synthesized silver nanoparticles (AgNP) (100–400 µg) incorporated was developed by Mohamed and Madian [85] to obtain improved mechanical and antimicrobial properties. The authors indicate that the mechanical properties of the CS film with AgNp (300, 400 μg) showed improved mechanical properties over the CS film, so the incorporation of a specific agent could improve the mechanical properties of the neat CS. Traction tests—developed by Almeida et al. [86]—were carried out for the mechanical characterization of the films tensile strength, elongation at break, elastic modulus of films-based CS, sodium acetate (NaA), and galactomannan (Gal). The values of the *TS*, *EAB*, and *EM* indicated that the films with Gal show lower values of *EAB* regarding CS and CS-NaA films, meanwhile with the addition of NaA had a lower *EAB* value (CS-Gal-NaA: 67.11 ± 0.89%), showing the statistical difference from the other films with *p* < 0.05. They observed that the most resistant packaging is the least flexible, showing lower *EAB* values. CS films exhibited the highest values of *EAB* and the lowest *TS* values. When NaA compound was incorporated a decrease in the value of *TS* of CS films (*p* > 0.05) was observed. The above “suggests that the incorporation of NaA compounds in the CS films generate to a decline in the intermolecular interactions” of the films. In this way, the crystallinity of the polymeric material is a parameter that must be taken into account since, an increase in the crystallinity of films can be associated with elastic with higher values of *EAB* and more breakable with lower values of *TS* films. Thus, the *EM* is a signal of the stiffness of the material, therefore, the larger the modulus, the more rigid the material.

Another analysis of mechanical properties carried out under the standard ASTM (American Society for Testing and Materials) D882 method through a testing machine shows that concerning the stress-strain curve of chitosan-based films with added gelatin (ge) and Quercetin/starch (Q). In this study, the mechanical properties of CS-based films were notoriously changed. *TS* value of the pure chitosan film was 10.54 ± 0.0565 MPa. With the addition of ge and Q, *TS* increased to 16.10 ± 0.1414 and 17.11 ± 0.3464 MPa, respectively. They found that higher *TS* in chitosan-based films could be due to better affinity of mixed CS and ge/Q. The increase observe of TS could also be related with the increased structural interaction between CS and Q. Besides, after the incorporation of ge/Q into the CS film, the *EAB* of the chitosan films reduced notoriously. The *EAB* of pure CS film was 11.04 ± 0.0565%, which decreased to 9.34 ± 0.4242 for CS-ge film and 5.100 ± 0.3162% for CS-ge-Q film. This could be due to the increased stiffness of CS-based films, considering that the formation of microspores and holes in the films because of the incorporation of Q which is observed from the superficial analysis [87]. The results of the incorporation of active agents, such as essential oils (e.g., eugenol (EU)), into the CS film on the mechanical properties have been studied by Zheng et al. [88]. They investigated *TS* and *EAB* for CS, CS-Acorn Starch (AS), and CS-AS-EU films. When CS-AS films were measured, they found that the *TS* values were higher regarding CS film. This increase in *TS* values of the film would be related to the intermolecular forces generated like hydrogen bonds between the NH_3_^+^ of CS and the OH^−^ of AS, thus enhancing the density of the final material. The *EAB* parameter value of the films were markedly changed due to incorporation of the AS. On the other hand, the *EAB* value partially decreased (68.5% to 60.4%) as the mass ratio of AS to CS increased by 0.9, showing a marked decrease in *EAB* and a further increase in AS contents.

Despite, the fact that different initiatives propose to develop biodegradable plastics, the mechanical and/or transport properties of these materials are parameters that still need to be improved to replace conventional plastic packaging materials and dispense with them entirely [89].

### 2.2. Barrier Property

In general, the barrier properties are defined as a module used to study the performance of films and/or packaging in food preservation. Among the barrier properties are oxygen barrier, water vapor, and light barrier property, the first two being the most relevant. The above are closely related to processes such as oxidation, microbial growth and food spoilage [90].

#### 2.2.1. Water Vapor Permeability (WVP)

The prevention of transfer moisture between food and the surrounding atmosphere is one of the main goals of food packaging materials. In this sense, the water vapor permeability (WVP) value must be very low [91]. Water vapor from the surrounding atmosphere can transfer to packaged foods, threatening the shelf life of foods by affecting quality and causing spoilage. The transfer of water either from or to the packaged food can negatively affect food products, generating changes at the structural and/or sensory level of these. *WVP* quantifies the amount of water that diffuses through the film per unit area and time (g/s m Pa), depending on the thickness and the differential pressure of the film [92]. *WVP* is calculated according to the combined Fick and Henry laws for gas diffusion through films, using Equation (4):(4)$$WVP=\frac{\Delta w\cdot x}{\Delta t\cdot A\cdot \Delta P}$$
where Δw/Δt $\left(g/s\right)$ is the flux measured as weight loss of the cell per unit of time, *x* (m) is the film thickness, $A\;\left(m^{2}\right)$ is the exposed area, and $\Delta P\;\left(Pa\right)$ is the water vapor-pressure deficit. *WVP* values are obtained gravimetrically following the ASTM Standard Test Method E96 [93].

Lyn et al. [94] elaborated CS films with the incorporation of graphene oxide (GO). They observed that for neat CS films, *WVP* decreased slightly (*p* ≥ 0.05) with the increasing heating temperature. This could be due to thermal crosslinking within the CS matrix, which increases tortuosity in the CS matrix. In the films with GO incorporated to CS matrix (CSGO) *WVP* decreased (*p* < 0.05). The researchers observed that temperature is a factor to consider, since they subjected the CSGO films to a thermal sweep between 60 and 120 °C and that the values obtained from *WVP* were significantly lower (*p* < 0.05) than when the films they were heated to 30 °C. The authors indicate that the findings of other authors have found similar results and attribute the decrease in WVP to a denser film matrix due to intense structural collapse when the film was heated to a high temperature.

In another work, Lin et al. [92] showed that the moisture permeability of composite films based on chitosan/okra powder/nano-silicon airgel compound increased slightly with increasing content of CS and nano-silicon airgel, but with a low difference. Based on its chemical structure, CS showed a better interaction with water molecules and aids in the transport of water vapor molecules through the film. At the same time, they observed a better desorption and adsorption of H_2_O molecules due to the incorporation of glycerol, which acts as a plasticizer. Consequently, due to the increase in CS, the hydrophilicity of the final material increased, which caused the dissolution of a significant part of the water vapor in the film and, therefore, the increase of the *WVP* value.

#### 2.2.2. Oxygen Transmission Water (OTR)

Oxygen causes a change in the quality of food if it enters the inner package. Oxygen permeability or oxygen transmission rate can measure the oxygen barrier characteristics of films: where the permeability or transmission higher, the poorer oxygen barrier property. Low oxygen transmission water (OTR) values indicates quite good oxygen barrier properties. Regarding CS-based films, it has been found that these have similar oxygen permeability values to the commercially available food packaging ethylene-vinyl alcohol copolymer films or polyvinylidene chloride films [90]. Oxygen permeability (OP) of films is determined using an automated oxygen permeability testing machine following the standard method (ASTM D3985-05, 2005). One side of the film sample is oxygen, and the other side is nitrogen. OTR and OP are calculated according to Equation (5):(5)$$OP=OTR\;\times \;\frac{d}{\Delta P^{*}}$$
where *d* is film thickness (mm), and Δ*P** is the partial pressure of oxygen [95].

Narasagoudr et al. [96] developed active packaging films based on CS, poly(vinyl alcohol) (PVA), and boswellic acid (BA). They evaluated the effect of BA compound on several properties of CS/PVA (CPBA) based active films. In this study, the *OTR* values of the CS film was 1197, film poly(vinyl alcohol) (PVA) 480, and CS/PVA film was 691 cc/(m^2^·d·atm). The *OTR* of the CPBA films ranged from 219.70 to 245.53 cc/(m^2^·d·atm). They observed that the incorporation of BA into the CS/PVA matrix significantly decreased the *OTR* values of the CPBA active films. The described fact could be attributed to the hydrogen bonds present in the final material. This study demonstrates that in the case of CS-based films, the barrier properties can be improved with the incorporation of various components chemically compatible with the CS polymer matrix.

### 2.3. Optical Properties

Among the optical properties, color and opacity are considered. Both are important parameters in the appearance of packaged foods since they affect the degree of acceptance by the consumer [97]. In the case of color, the luminosity values (L*) and chromatic coordinates (a* and b*) of the film are evaluated by reflectance measurements. Here, L* ranging from black to white, and the horizontal axes, indicated by a* and b*, ranging from −a*: greenness, −b*: blueness, to +a*: redness, +b*: yellowness. The values of a* and b* approach zero for neutral colors and increase as the color becomes more chromatic and more saturated [86].

Color difference $\left(\Delta E^{*}\right)$ is calculated by Equation (6):(6)$$\Delta E^{*}=\sqrt{{\left(\Delta L^{*}\right)}^{2}+{\left(\Delta a^{*\;}\right)}^{2}+{\left(\Delta b^{*}\right)}^{2}}$$
where: ΔL* = Lstandard∗ − Lfilm∗; Δa = astandard∗ − afilm∗; and Δb∗ = bstandard∗ − bfilm∗ [97].

In a study, a CS film based was developed, where zinc oxide (ZnO) particles (0, 1, and 3% (*W*/*V*)) and Melissa essential oil (MEO) (0, 0.25, and 0.5% (*W*/*V*)) were used with the aim of improving the biopolymer properties. The film color study using a colorimeter showed that pure CS film has a clear, yellowish appearance. Meanwhile, the composite films exhibited a great difference when compared with the pure CS films. With the increase of the concentrations of Melissa essential oils and zinc oxide they observed that the brightness parameter L* decreased. The researchers concluded that this result is due to the phenolic compounds present in the films which have the spectroscopic characteristic that can absorb light at low wavelengths. In a study that used the same *EO* in a mucilage film, similar results were obtained. They observed that with increasing concentrations of MEO and zinc oxide nanoparticles, the value of a* increases, and the color of the samples tends to be red. The color of the films tends to yellow when the concentration of MEO increases. On the other hand, the films with high concentrations of zinc oxide results clouded. This result can be due to the agglomeration of zinc oxide Np in the CS films [98].

Reflectance measurements are used to describe the opacity of a material. Besides, opacity allows measuring light transmittance of the films, through an ultraviolet–visible (UV–Vis) spectrometer in the wavelength range of 200–800 nm. Film opacity is determined by measuring absorbance at 600 nm and calculated by the Equation (7):(7)$$\mathrm{Opacity}=\frac{A_{600}}{t}$$
where A_600_ is the absorbance at 600 nm wavelength and t is the thickness of the film (mm) [99].

Xu et al. [100] developed CS-gum arabic-based polyelectrolyte films complexed with cinnamon essential oil (CEO) and clove essential oil (CLO). They analysed the opacity, observing an increase from 0.71 to 2.53 when the CEO increased (0% to 15%) in the polymer matrix. These results could be attributed to the increased intensity of light-scattering induced by the increasing size of CEO and CLO droplets within the film matrix, therefore, the changes in the opacity parameter are due to the increase in oil in the polymer matrix. Regarding films containing CLO, they resulted in the most transparent. While the addition of CEO increased the yellowish color of control films, which interfered in of Schiff-base reaction between CS and CEO. They concluded that this organic reaction between CS and CEO would help the chemical compatibility of the CS/CEO film. Base opacity of CS films can increase or decrease depending on the active agent that is incorporated into the matrix of the polymer and the greater or lesser affinity they have with it. Both color and opacity of food containers are relevant factors for the choice of consumers. In addition, the opacity parameter plays an important role in sensitive to light foods.

### 2.4. Antimicrobial Activity

CS is biopolymer that has outstanding antimicrobial activity against several microorganisms such as bacteria (Gram-negative and Gram-positive), filamentous fungi, and yeast [101,102]. Many studies have investigated and demonstrated the antimicrobial properties of CS, however, the mechanism of action is still diffuse [103]. In contrast, the antibacterial activity is a complex process that most of presents differences in effectiveness between Gram positive and Gram-negative bacteria due to the different characteristics of the cell surface [104]. Rapa et al. [105] studied the effect of the biopolymer CS concentration on the properties of films containing poly (lactic acid) (PLA) plasticized with tributyl *O*-acetyl citrate (ATBC). The samples were arranged as films and sheets (PLA/CS and PLA/ATBC and samples of neat PLA) to evaluate the changes in their properties and in the antimicrobial activity of these, especially with respect to the antifungal and antibacterial activity. The antifungal capacity was evaluated in Petri dishes with potato dextrose agar (PDA), directly exposing the samples and evaluating inhibition halos and in the case of antibacterial analyzes, they were carried out based on the ISO 22196 standard.

In the case of the antifungal analysis, they evaluated the inhibition of the growth of different fungi such as *Aspergillus brasiliensis*, *Fusarium graminearum* and *Penicillium corylophilum* against the samples. The results regarding the inhibition rate (% IR) at the contact surface were expressed as a percentage. % IR against the fungus *Aspergillus brasiliensis* varied between 99% and 100% in the contact surface for both samples composed of PLA/CS, in the form of films and sheets, and they were compared with samples without CS. In all the cases analyzed, the % IR values were higher for the sheets than for the film samples. They observed that with the increase in the amount of CS in the films, the % IR values increased from 99.91% to 100%.

On the other hand, % IR against the fungus *Fusarium graminearum* were between 84.16% and 100% at the contact surface. By increasing the amount of CS in the mixture they observed that the value of % IR 84.16 increased to 99.41% and finally, for *Penicillium corylophilum* they obtained IR between 94.5% and 100%. Based on the results obtained it is possible to specify that both the amount of CS present in the mixture, as well as the form (film or sheet) influences the % IR against the microorganisms tested. In the case of antibacterial analysis, they analyzed the effect of PLA samples with chitosan on *Staphylococcus aureus* and *Escherichia coli*. Samples (films and sheets) containing 1% or 3% CS showed a potent antibacterial effect, thus reducing approximately 2.7–2.8 log units compared to control or untreated PLA. In this case, the effects were diminished to values of 1 log unit when increasing the CS to 5%. The antimicrobial effect observed for *E. coli* was greater than for *S. aureus*. All samples containing CS showed excellent antimicrobial effect [106,107,108,109]. This study shows, on the one hand, that CS would be effective against more than one type of microorganism, filamentous fungi, as well as bacteria, but in the case of bacteria a greater activity against Gram-negative.

### 2.5. Heat Resistence 

Commonly, thermal properties of polymers are studied through differential scanning calorimetry (DSC) and thermogravimetric analysis (TGA) [110]. In food packaging it is very important to quantify these properties with the aim of knowing the thermal stability of the materials that will be part of the food packaging system [111]. For thermal analysis, integrated DSC and TGA equipment or independent equipment can be used. Aguirre-Loredo et al. [112] studied glass transition temperature (Tg), related to the plasticization in biopolymers, such as CS. Tg indicates the temperature region at which a polymer changes from a solid state to a viscous state. In food packaging, changes in Tg is a pretty important parameter that can affect the quality of the product. They found that the glass transition temperature can change depending upon the composition and moisture content of the material.

Meng et al. [113] analyzed the thermal properties of starch-chitosan based films through DSC. First, a starch/chitosan film was observed showing the classic endothermic peak (95.81 °C) due to loss of water. Then, in the case of starch/chitosan films with peanut shell (in three different concentrations), for each of them they reported an endothermic peak with a higher temperature area, whose values fluctuate between 108.33 °C, 140.90 °C and 150.93 °C, respectively, as the concentration of the peanut shell in the starch/chitosan matrix increased. This result indicates that films with incorporated peanut shell extract are more thermally stable. However, different results were obtained when peanut skin extracts were incorporated into starch/chitosan films. As the concentration of this extract increased the endothermic peak decreased. The investigators indicated that this behavior can be explained through hydrogen bonding which is destroyed in the film matrix due to the high content of peanut skin.

Kizilkonka et al. [58] developed chitosan-based composite films with the incorporation of cerium oxide nanoparticles (NPs). They studied the thermal behavior of the films through TGA measures finding two main weight loss regions of the films. Thermal decomposition was observed between 180.24 and 460.97 °C, and this was attributed to dehydration of the saccharide rings, depolymerization, and decomposition of the acetylated and deacetylated units of the CS. 

In a study by Koc et al. [114], they used a mushroom extract from *Tricholoma terreum* (edible mushrooms) to produce CS-based films. They studied a series of variables and parameters between those jointly studied by DSC (Figure 4A) and TGA (Figure 4B). In the DSC analysis, they observed the glass transition temperatures of both pure CS films and CS films with antifungal extract. In Figure 4A, it is possible to observe endothermic peaks for both films at 74 and 108 °C respectively. These thermal events could be attributed to the loss of water from the films in the free state. Then other thermal events were observed linked to the glass transition temperature of CS at approximately 150 °C, and in the case of the CS film with antifungal extract this value was increased (195 °C). These variations could be explained based on the increase in the number of hydroxyl groups forming intermolecular hydrogen bonds. At the same time, it was possible to observe a crystallization phenomenon of the films of pure CS and of CS with extract around 273 and 302 °C, respectively.

In the case of TGA (Figure 4B) for the same samples mentioned above, three mass drops were observed for the pure CS film and four in the case of the film with extract. In both cases the first degradation was associated with unbound water (30–100 °C), the second drop in mass was associated with the degradation of the plasticizer used 157 °C and 199 °C respectively, a third degradation was associated with the decomposition of CS (266 °C and 257 °C), a value that varies according to the source of CS and other factors additional to its processing. A fourth mass loss was observed only in those films that contained the extract (above 270 °C).

DSC and TGA were used to study the effect of agents on the stability of composites. The thermal degradation of the films was changed to higher temperatures with the incorporation of components, this indicates that adding components (such as nanoparticles, essential oils, other polymers, etc.) into the film matrix generates a thermal stabilization of the final material [114,115].

### 2.6. Biodegradation

Eco-friendly materials have become preferred alternatives both by the industry and consumers as part of food packaging, reducing the use of materials derived from fossil fuels. This is because it seeks to reduce the environmental impact of the tons of waste that are generated in the world [116]. The biodegradation process can be in accordance with ASTMD5338-15 [117]. Deshmukh et al. [99] prepared and characterized films based on CS, plasticizer (glycerol), and defatted *Chlorella* biomass (DCB) in terms of biodegradability.

They observed that the films initial structural integrity was missing. The films showed holes and loss of shape. The above indicates that the films react with the soil. The authors confirmed the structural and morphological changes by scanning electron microscopy (SEM) analysis. Surface degradation of pure chitosan film and CS/25% DCB film was found before (0 days) and after 60 days of the test of being in contact with the soil. This indicates that there was a degradation of the films caused by microbes in the soil. The total weight loss of the samples is measured after the incubation period using Equation (8):(8)$$\mathrm{Weight}\;\mathrm{Loss}\;\left(\;\%\;\right)=\frac{W_{1-\;}W_{0}}{W_{0}}\times 100$$
where $W_{1}$ is the final weight of the specimens after incubation and $W_{0}$ is the initial weight of the specimens before incubation [118].

Zhu et al. [119] investigated the characteristics of polyelectrolyte complex (PEC) of chitosan and sodium cellulose sulfate (NaCS), focusing on the biodegradeability of CS, NaCS, and chitosan/NaCS PEC films with pepsin, trypsin, lipase, alpha-amylase, and cellulase. The authors mention that previous studies on the degradation of chitosan indicate that the lytic activity of CS and the kinetic parameters of enzymes from different sources (even those of origin) change. For example, *Aspergillus oryzae* lipase, exhibited relatively low chitosan lytic activity, but *Aspergillus niger* lipase showed chitosan lytic activity comparable to chitosanase. Therefore, the speed and effectiveness of the degradation process will depend on the environment where it is carried out. Gan et al. [120] developed the cross-linking of cellulose nanocrystals (CNC)/CS composite films with glutaraldehyde (GA) as the cross-linking agent. To evaluate the biodegradation process, they used solid state fermentation to determine the ability of fungal isolates (*Ophiocordyceps heteropoda*, *Enterobacter kobei* and *E. roggenkampii*) in the degradation of the films. The filamentous fungi were cultured on PDA agar for more than a week at 30 °C. When evaluating the weight losses of the uncross linked and crosslinked CS 2composite films, an approximately linear relationship with biodegradation time was observed. The greatest weight loss was presented by the uncross linked pure CS film, and it was approximately 72% after 15 days of being in the soil.

The study shows that the degradation rate decreases in the composite films that containing CNC. Other authors also studied the biodegradation of CS samples but in compost. After 7 days of incubation, the CS films show remarkable physical changes such as swelling and were completely covered with the white mycelial growth of the compost microorganisms. Over the next 15 days, the samples showed a rapid disintegration. However, CS films were able to disintegrate only after five weeks [121].

## 3. Starch

For some time, starch has been highlighted as a great alternative [10,12,30,37]. The reasons for the interest in using starch are its low cost, biocompatibility, non-toxicity, and easy chemical modification. Clearly, among these characteristics, their chemical structure is one of the most attractive, since offers a lot of possibilities to improve or incorporate certain properties according to each particular application [20,21,31,32]. As was reviewed in Section 1, starch has been used for different purposes, in many ways from blends to composites (Figure 5). Nowadays, given complex environmental problems is necessary to open the doors to the use of biopolymers and the properties of starch make it an excellent alternative.

### 3.1. Starch as Pollutants Remover

Nowadays, the scarcity of water is a problem that many industries must face. Unfortunately, industrial growth causes a huge discharge of polluted water, and much of this water is not treated before disposal. Industrial water contains pesticides, heavy metals, toxic dyes, organic and inorganic pollutants, and much more. In this context, heavy metals can cause serious damage to living organisms, dramatically affecting the ecosystem of various species. In animals and humans, they can produce the disruption of enzymatic functions resulting in serious health consequences [122].

The use of adsorption as a method for removing pollutants is attractive due to its low cost, high efficiency, and easy procedures. The use of starch—whether it is chemically modified, blended, or in composites—as an alternative contaminant removal material has been studied [123,124,125,126,127]. Various techniques are usually developed to characterize this type of systems. Among them are adsorption studies, Fourier transform infrared spectroscopy (FT-IR), X-ray diffraction (XRD), SEM, transmission electron microscopy (TEM), and adsorption kinetics. These characterizations definitely help to understand the performance of starch as an adsorbent agent. However, in recent years, there have appeared other characterizations that could offer a larger understanding regarding the physical properties of adsorbent agents. Below, we will review some of them.

### 3.2. Superficial Analysis

#### 3.2.1. Energy-Dispersive Spectroscopy (EDS)

The energy-dispersive spectroscopy technique is an experimental method that allows semi-quantitative measurement. This superficial technique is used as a complementary characterization of SEM, coupled according to sample requirements. In the study by Chen et al. [128], CaCO_3_ was added on the surface of starch-FBO (ferromanganese binary oxide) to form Ca-starch-FBO, aiming to improve the adsorption capacity of Cd(II) and the removal of As and Cd at specific pH conditions. The EDS results showed that the distribution of Ca, As, and Cd was similar. Fe and Mn that were used during the treatment also were recorded. The latter might affect the adsorption of the material of interest. A ternary complex could be formed, which would affect the adsorption capacity due to competition for active sites in the adsorbent material. FT-IR corroborates this assumption.

Cellulose nanofibers (CNFs) can be used as adsorbent material. Baghbadorani et al. [15] treated cellulose nanofibers on starch-g-poly(acrylic acid) (St-g-PAA) for Cu^2+^ elimination in an aqueous medium. This composite improved the adsorption of ions in the study. Characteristic such as the negative charge of CNFs and their large surface area allowed the composite to reach its goal. EDS characterization was performed to confirm Cu^2+^ removal. They observed that there is 52.89 wt% on the surface of the composite prepared (Figure 6). 

In another study about Cu removal (II) from aqueous media, Dai et al. [129] elaborated a hydrogel containing sodium alginate and 2-hydroxy-3-isopropoxypropyl starch. Once the hydrogel and Cu(II) were in contact, they used EDS characterization to verify the presence of Cu(II) in the hydrogel surface. The results of this technique showed that Cu(II) was successfully adsorbed by the hydrogel prepared. They also observed that the hydrogel maintained its signal after the adsorption of Cu(II). This indicates a good stability of the hydrogel used. A novel material constituted by stable starch with loaded nano zero-valent iron was developed by Yang et al. [130] to remove Cr(VI) from wastewater. They used the EDS method for studying Fe distribution. In this sense, they observed that the distribution of Fe onto stable starch was uniform.

#### 3.2.2. X-ray Photoelectron Spectroscopy (XPS)

X-ray photoelectron spectroscopy (XPS) is a compound analysis quantitative technique that measures surface elemental composition. XPS can be useful for adsorbent characterizations as it might clarify the adsorption mechanisms involved. Mittal et al. [131] developed a new magnetic adsorbent using Fe_3_O_4_ nanoparticles and starch as a carbon source. The deposition of nanoparticles, Fe_3_O_4_, onto the surface of carbonized starch was confirmed through XPS analysis. Hg^2+^ is a long-lasting toxic metal that causes serious damage by accumulating both in the environment and in humans. Human activities in industrial areas are responsible for this pollutant. In this regard, Naushad et al. [132] developed starch/SnO_2_ nanocomposites for the removal of Hg^2+^ from aqueous media. Through XPS characterization, they found that the adsorption of Hg^2+^ was carried out at superficial level, meaning that Hg^2+^ was adsorbed onto the nanocomposites. They also verified that Hg^2+^ did not change its oxidation state.

On the other hand, mechanisms involved in adsorption can also be elucidated using the XPS technique. In this way, Fu et al. [133] developed an adsorbent from a starch-based polysulfide complex material with the aim of capturing Hg(II) and then converting the absorbent material into a catalyst. Through XPS, they proposed a possible mechanism of adsorption. Firstly, XPS showed that Hg(II) was successfully adsorbed. Then, they observed that O, N, and S are involved in Hg adsorption. The researchers proposed that several factors are involved in the adsorption of mercury such as metal chelation, complexation, and electrostatic attraction. Figure 7 details the above.

The adsorption of gold becomes interesting since this element is a non-renewable precious resource. This metal is used in several applications, mainly for electronic and catalysis applications. In this sense, there is a large amount of electronic waste containing gold that causes serious damage to the environment. Thus, Liu et al. [134] prepared a composite through a crosslinking reaction between tannin acid and dialdehyde corn starch to be used as bio-adsorbent and remove Au(III) from wastewater. The XPS technique was used to try to elucidate the mechanism of Au adsorption (III) on starch-based composite. They found that when the adsorbent was added into the Au(III) solution, electrostatic interactions were produced. Then, a redox reaction was carried out where hydroxyl groups of adsorbent were oxidized to carbonyl groups, while Au(III) was reduced to Au(0). The latter could be observed since gold particles were detected in the bio-adsorbent surface.

Xue et al. [135] developed a film consisting of CdS and carboxymethyl starch with the aim of removing certain dyes from water. In this case, they used XPS to establish what type of interaction was between CdS and modified starch (Figure 8). In this sense, they observed that CdS interacts with carboxymethyl starch (CMS) through electrostatic interactions. There is no chemical bonding involved. Based on this literature, it can be observed that the use of the XPS technique is a great complement to both verify the adsorption of the substance of interest and elucidate the adsorption mechanisms involved.

#### 3.2.3. BET (Brunauer–Emmett–Teller)

BET (Brunauer–Emmett–Teller) theory is a method used to study the adsorption phenomena on solid surfaces. Although BET is not a new technique, its use as a complementary technique for studying pollutant adsorbent materials has increased in recent years. It certainly is a technique that should be included in the characterization of this type of material since, through this characterization, it is possible to know the porosity type of the adsorbent material and thus achieve an appropriate application. Mittal el al. [131] developed a magnetic carbonaceous adsorbent from corn starch. BET studies showed wide distribution of pore size generated in the adsorbent surface. In this study, it was observed that the magnetization involved improved the porosity with a higher surface area. Pore volume was improved too. Therefore, BET characterization allowed the observation that the porous structure of corn starch was enhanced with the incorporation of Fe_3_O_4_ nanoparticles for the adsorption of cationic dye from wastewater. In another work, Stan et al. [136] modified the starch surface with Fe_3_O_4_ magnetic nanoparticles prepared through an environmentally friendly synthetic route. In this work, BET results showed a higher surface area in starch with nanoparticles when compared to starch without nanoparticles. The type of porous corresponded to mesopores and the average pore size radius was 12.55 nm.

Furthermore, Tao et al. [123] modified starch with ZnMgAl-LDHs (LDHs: Layered double hydroxides) to improve the adsorption properties of starch. They observed that the surface area had increased, however, the excess of starch generated a collapse in the hydrotalcite structure resulting in the reduction of adsorption capacity of the material. Priyanka and Saravanakumar [124] designed a novel starch-derived zinc-carbon foam-like (Zn-CFst) material to adsorb different dyes. BET analysis indicated a highly mesoporous surface with an average pore diameter of 31 Å. The adsorbent efficacy markedly improved due to the pore diameter and mesoporous surface. These are both quite important for quick and efficient dye adsorption.

### 3.3. Magnetic Properties

The use of magnetic material such as nanoparticles (for instance, Fe_3_O_4_) is steadily increasing in the world of starch-based adsorbents. The purpose of this is to stabilize the tendency to aggregation of nanoparticles through the development of biopolymer coatings. In this sense, the vibrating sample magnetometer (VSM) method was used to measure the electromagnetic properties of different materials containing magnetic structures. This characterization can provide important information regarding the stability of the magnetic properties of material incorporated into starch.

In the previous section, we reviewed the work by Stan et al., [136] who developed a green remover to remove dyes using starch coated magnetic nanoparticles. Through VSM measurements, they observed that the saturation magnetization values decreased regarding non-stabilized Fe_3_O_4_. This result could be attributed to the starch coat. Meanwhile, lower saturation magnetization values were observed after the adsorption of dye. Aiming for water remediation, Perez et al. [137] developed a material based on manganese ferrite (MnFe_2_O_4_) nanoparticles covered with starch. Magnetization measurements of the starch-coated MnFe_2_O_4_ showed a decrease in saturation magnetization regarding MnFe_2_O_4_ nanoparticles, which could be due to functionalization with starch.

Starch-stabilized zero-valent iron nanoparticles were synthesized by Okuo et al. [138] with the purpose of being used as potential synthetic immobilizing agents in the remediation of soils contaminated with Pb, Cr, Ni, and Cd. VSM analysis showed that starch-stabilized zero-valent iron nanoparticles were superparamagnetic. This characteristic indicates the great stability of this material, both thermally and chemically.

Gong et al. [139] developed magnetite (Fe_3_O_4_) nanoparticles using starch as stabilizer for the removal of aqueous perfluorooctanoic acid (PFOA). This acid is widely used in manufacturing industry. Unfortunately, it is a species listed as persistent, being detected both in the environment and human beings. PFOA shows bioaccumulation, toxicity, and resistance to typical environmental degradation processes. The magnetic properties of non-stabilized and starch-stabilized magnetite were measured. The saturation magnetization value was lower in starch-stabilized Fe_3_O_4_ than in non-stabilized Fe_3_O_4_. They attributed this difference to the size effects. The surface/volume rate of starch-stabilized nanoparticles is large, which could have structural effects resulting in magnetic differences between starch-stabilized Fe_3_O_4_ and non-stabilized Fe_3_O_4_.

Another study worked with Fe_3_O_4_ nanoparticles using starch-salicylaldehyde based nanocomposites and salicylaldehyde resin as stabilizer for their potential application as Pb(II) and Cd(II) adsorbents in aqueous solutions [122]. VSM characterization showed, as we have seen previously, that the saturation magnetization value in stabilized-Fe_3_O_4_ nanoparticles is lower than in non-stabilized-Fe_3_O_4_ nanoparticles. These results indicate that the magnetic properties in stabilized-Fe_3_O_4_ nanoparticles nanocomposites are still valid and can be removed from the matrix after the adsorption process. The use of magnetic adsorbents for the removal of aqueous pollutants has gained attention as it allows for their easy separation through the application of a magnetic field. In general, in the studies reviewed, it can be observed that the magnetic properties of the nanoparticles used were maintained.

### 3.4. Molecular Dynamics (MD) Simulation

Molecular dynamics is a computational simulation technique developed through approximations of known physics in which both atoms and molecules interact over a period, giving a vision of the movement of the particles. These simulations are frequently used in research related to proteins and biomolecules, as well as other materials studied by science.

One of the investigations in this area is the work carried out by Mofradnia et al. [140], who studied the efficacy of starch-modified zero-valence iron nanostructure for nitrate removal. They simulated the effect of zero-valence iron/starch nanoparticles in contact with *Thiobacillus dinitrificans* for the removal of nitrates, using materials study software, based on thermodynamic principles and appropriate equations, through a simulation of molecular dynamics.

For the optimization of the work, the structure of *Thiobacillus denitrificans*, they suggest extracting it from the RCSB (Research Collaboratory for Structural Bioinformatics) database, doi:10.2210/pdb3sb1/pdb. On the other hand, nitrate structures and starch molecules they since from the PubChem molecular bank. Subsequently, they added starch molecules to zero valence iron nanoparticles with the intention of increasing the stability of the adsorbent and preventing the oxidation of the nanoparticles. With this they found that the starch fulfils a membrane function for the surface of the iron nanoparticles, also protecting the structure of the nanoparticles closest between the microorganism and the zero-valence iron nanostructure. This caused the reactivity of the nanoparticles to increase when starch molecules are added for the elimination of nitrates.

The structures designed them based on microorganisms in an aqueous medium, where the accumulation of its components was carried out through a module of amorphous cells tools and a compass force field at a constant pressure of 101.325 kPa and a temperature of 298 K. On the other hand, the percentage ratio was 45.3% nitrate and 54.7% water. In addition, they used the periodic boundary conditions (PBC) for the proper distribution of the components in the box and the unwanted parameters were optimized to 10,000 energy units.

For the final investigation of molecular dynamics, they used the Forcite module with the dynamic operating tool and the compass force field. To avoid computational interference, they suggest that the system be neutralized early in the process. The calculations were simulated at constant temperature using the NVT computational equation and the Nosé–Hoover equation (NHL), where they selected an adequate temperature control based on the presence of nanoparticles in the system. To condition the system to simulate the experimental conditions, they established that all parts of the system were at ambient pressure. Each of the steps were performed at 5 ps and the calculations at 1 fs intervals.

To present the results of the predictions made in this research, they conducted studies based on radial distribution (RDF), density, potential energy, and temperature. One of these results showed the simultaneous increase in nitrate removal efficiency when the zero-valence iron/starch nanoparticles and *Thiobacillus dinitrificans* were present, reaching 91%, compared to when there were no nanoparticles in the system, reaching only 44.44%, verifying that the removal of nitrates increased considerably with the presence of nanoparticles, also increasing the stability of the system. Then, the extracted results indicated that the density reached 1.65 g/cm^3^ in the presence of the nanoparticle, much higher than that calculated in the absence, reaching a value of 0.82 g/cm^3^.

Another work that used a molecular dynamics (MD) simulation was carried out by Cui et al. [141], where they simulated the relationships between the mixtures of starch, urea and water using GROMACS 5.1.4. They introduced a starch fragment containing 55 glucose residues, with structures similar to a helix, to simulate the real structure of starch and represent its characteristics and properties, saving computational resources.

On the other hand, the glucose force field was modified from the OPLSAA (Optimized Potentials for Liquid Simulations all atom) force field for carbohydrates, suitable for the model molecule [142]. To carry out the study and have the appropriate coordinates, they requested the molecular model and the force field from the Shanghai Institute of Technology (Shanghai, China). As for the pdb (Protein Data Bank) file (Protein Data Bank), they use the urea topology file they were generated through the LigParGen server with OPLSAA force field [142,143,144,145], using the TIP4P (Transferable intermolecular potential with four points) water model. In all simulations, the molecular model was incorporated in the center of a 9 × 9 × 9 nm^3^ cubic box with periodic boundary, where the system contained 0, 900, 1900, 2900 and 3900 urea molecules. They set the Lennard–Jones potential truncation radius to 1 nm, and constrained all link lengths using the LINCS (Linear Constraint Solver) algorithm.

Regarding the short-range electrostatic interaction, in addition to the van der Waals interaction, these were handled at 1 nm, and to calculate the long-range interaction the Ewald particle mesh method (PME) was adopted. Regarding the temperature, they set it at 300 K and the pressure at 1 atm using the Parrinello-Rahman barostat [146]. To observe the positions, velocities and energies, a record was made every 10.0 ps.

The most important results of this research are the significant change experienced by the structure and morphology, as well as the thermal property of starch in the presence of urea, showing that it depends positively on the urea content of aqueous solutions.

Regarding the molecular dynamics’ simulations, the study focused on the interaction of hydrogen bonds, in the distribution of starch, urea and water fragments, both at the molecular and atomic level. Noting that the hydroxyl oxygen atoms in the glucose residue, the nitrogen atoms in the urea molecule and as well as the oxygen atom present in the water molecule are the main hydrogen bonding sites between the solute and the solvent.

On the other hand, they also observed that urea molecules replaced the place of certain water molecules due to the formation of hydrogen bonds with the hydroxyl oxygen atoms of the starch, altering the water network around the starch molecule. Therefore, this study provided important information on the phase transitions of starch in aqueous medium. They concluded that this simulation could potentially be used to design starch-containing materials.

### 3.5. Electronic Structure

Another potential use of computational studies is through functional density theory (DFT), which is applied to electronic systems. This an alternative variational procedure to the solution of the Schrödinger equation, where the functionality of electronic energy is minimized with respect to electronic density. This method is currently one of the most widely used for work on quantum calculations of the electronic structure of matter, especially in research on the physics of condensed matter, as well as quantum chemistry.

One of the works that used this method was carried out by Bashir et al. [147], who synthesized a polymer of potato starch phosphate (PSP) using ultrasound microwaves, which they carried out by crosslinking the hydroxyl groups of native potato starch (NPS) and using phosphoryl chloride as a crosslinking agent. In this theoretical-experimental investigation, a study was carried out on the selectivity of metal ions, and the sorption mechanism.

To validate the experimental selectivity and further investigate the chemical interactions between the Zn(II), Cd(II), Pb(II) and Hg(II) ions and the biosorbent PSP, DFT incorporated in the Gaussian code set 03 [148]. To optimize the geometries of the molecules they used Becke’s three-parameter hybrid model, using the Lee-Yang-Parr (B3LYP) and LanL2DZ set of functional correlation bases [149,150]. To simulate chemical selectivity, they used a cross-linked starch fragment as a model for computational details. The particularity is that in the chosen fragment, there were abundant hydroxyl groups available for the attachment of metal ions, including surface hydroxyls, glycosidic hydroxyls, and phosphate hydroxyls.

To perform the load study, these researchers performed a Mulliken load analysis, which allows the measurement of elemental load. As a result, they observed that oxygen phosphate atoms carry more negative partial charges than terminal oxygen or glycosidic oxygen, suggesting that oxygen phosphate is a more favourable site for adsorption of metal cations. The above allowed them to optimize the geometries of the PSP–M^2+^ cluster models, where the cation, M^2+^ (M = Zn, Pb, Cd and Hg), binds to the oxygen phosphate atoms in an optimized structure. On the other hand, the binding energy values (Ebd) were calculated as Ebd, where the positive values obtained indicated that the adsorption was favourable and the interaction was stable [151].

Another important step was the calculation that they made to the binding energy values for the Zn(II), Cd(II), Pb(II) and Hg(II) ions. The DFT calculations showed that PSP exhibits the strongest adsorption selectivity towards Zn(II) among all the heavy metal ions tested. This result agreed with the paradigm of Pearson’s HSAB (Hard soft acids bases) principle, where it is estimated that the border “cations Zn(II) and Pb(II) interact more strongly with the hard anion (O^−^) and, consequently, the soft cations (Cd(II)) and Hg(II)) interact less [149]. These quantum calculations using density functional theory (DFT) supported experimental results of adsorption selectivity that Zn(II) > Pb(II) > Cd(II) > Hg(II), in terms of bond energy measurements metal-oxygen.

### 3.6. Reusability

The recyclability of the adsorbent is quite an important factor for it to have feasible applications. In this sense, the stability of the adsorbent is key to achieve this property, and the total cost of the adsorbent will be an important advantage. Adsorbent reusability is absolutely necessary in the recycling context. The fact that a material can be used more than once causes a positive impact on the environment. On the subject of the physical properties of adsorbents, this characterization can provide important information about the conformation of the material and its stability.

Through reusability tests, Ahamad et al. [122] showed that their adsorbent could be reused until 5 cycles regarding Pb(II) and Cd(II) adsorption (Figure 9). After seven cycles, there was a slow decrease in the adsorption capacity of these metals. In the previous section, we saw that they developed this adsorbent using starch-salicylaldehyde nanocomposites. Similarly, in 2020, Perez et al. [137] presented an adsorbent based on magnetite nanoparticles (Fe_3_O_4_) stabilized with carboxymethylated starch. Reusability was observed even after 4 cycles of Pb(II) removal.

Mittal et al. [131] introduced an adsorbent prepared with magnetized and starch-based carbonaceous material. This material was used to study methyl blue removal. Herein, the results of reusability tests showed that, throughout six cycles of adsorption-desorption, there was good efficiency of above 90%. Figure 10 shows the reusability test performed by Suo et al. [152]. They developed a mesoporous activated carbon adsorbent from starch to remove pesticides from water. They found that, after five consecutive cycles of adsorption–desorption, the adsorption efficiency of 11 pesticides was over 80%.

In general, an excellent adsorption efficiency is observed in starch-based materials. This characterization indicates great chemical and physical stability, exhibiting high efficiency in up to five cycles of adsorption.

## 4. Overview

The use of biopolymers in several areas, such as the replacement of conventional polymers, is a challenge. Chitosan and starch present attractive physical and chemical properties, which make them excellent alternatives for this change. The use of these biopolymers, either through the mixing with other polymers, the formation of composites, or the chemical modification, allows an improved material with the desired properties to be obtained.

In the food packaging industry, chitosan is an excellent candidate to replace conventional polymers because it exhibits a great capability to form films with suitable mechanical properties such as stress-resistant, flexibility, and elasticity properties, and on the other hand, antimicrobial activity. The packaging industry attaches great importance to protecting the food and making it attractive for customers. In this sense, advancements in packaging ensure the quality of food products inside the packaging, regarding shelf life and usability, to the customers. Meanwhile, in the field of aqueous pollutant removal, starch is in high demand due to its characteristics, i.e., being biocompatible, affordable, and chemically attractive. In this review, we have identified some highlighted characterizations concerning chitosan in the food packaging field and starch in the pollutant removal field, such as mechanical, thermal, optical, barrier, biodegradability, reusability, molecular simulation electronic structure, superficial, and magnetic properties characterization. All these characterizations give us necessary and complementary information to better understand the properties of these biopolymers in the respective areas reviewed.

## Figures and Tables

**Figure 1 polymers-13-01737-f001:**
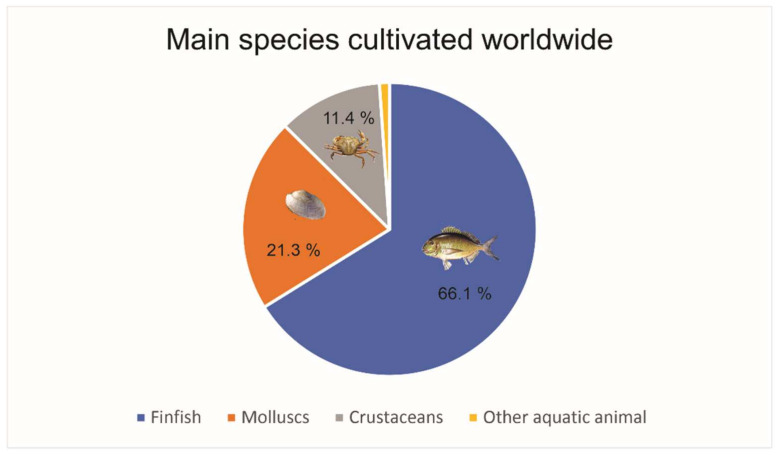
Main species cultivated worldwide in 2018 [4].

**Figure 2 polymers-13-01737-f002:**
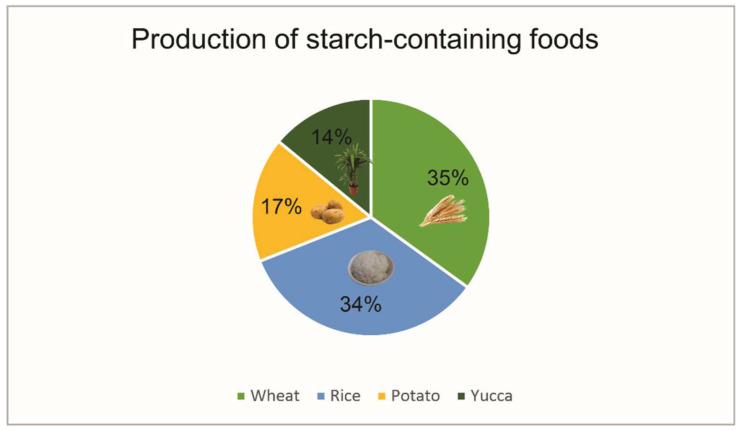
Starch-containing food products produced in 2019 [5].

**Figure 3 polymers-13-01737-f003:**
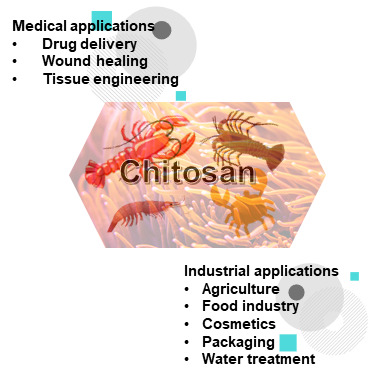
Some chitosan applications.

**Figure 4 polymers-13-01737-f004:**
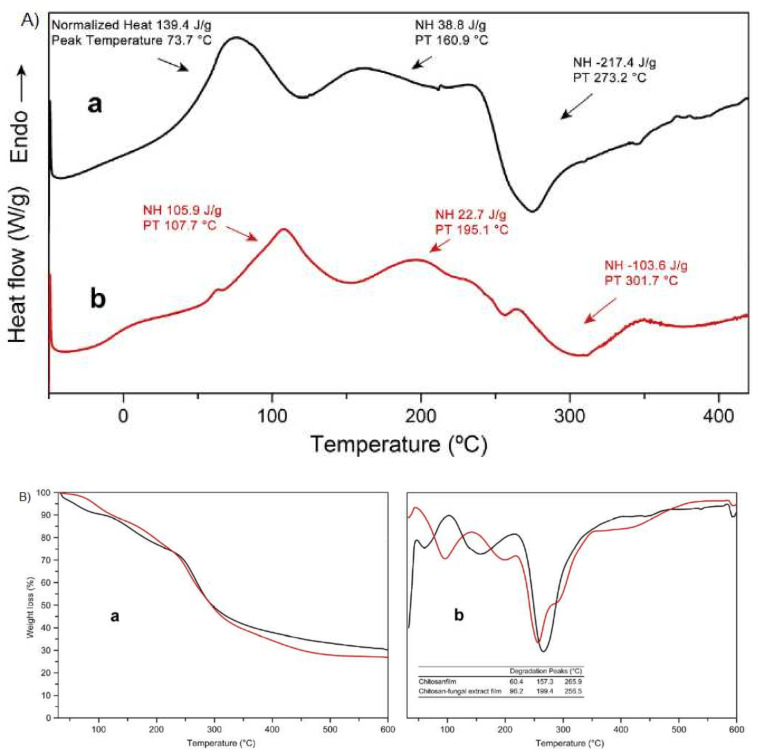
(**A**) Differential scanning calorimetry (DSC) thermograms of (**a**) chitosan and (**b**) chitosan with extract fungi incorporated in films; (**B**) (**a**) thermogravimetric analysis (TGA) and (**b**) DTGA (derivative thermogravimetric analysis) of chitosan and chitosan with extract fungi incorporated in films. (Reprinted from ref. [114]. Copyright 2020 with permission from Elsevier).

**Figure 5 polymers-13-01737-f005:**
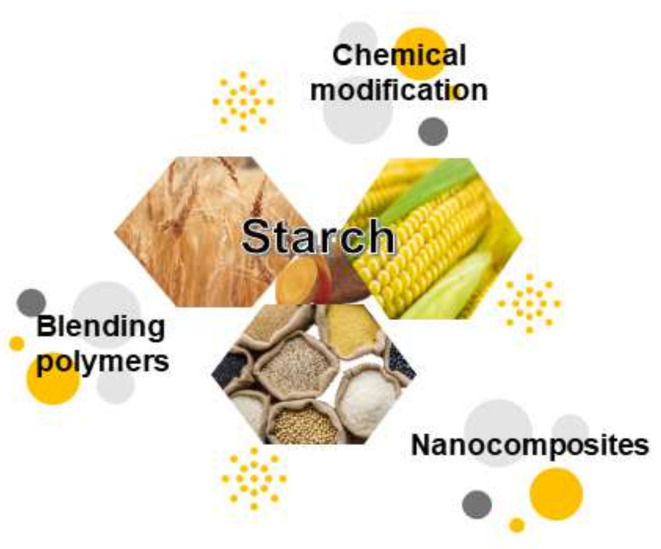
Use of starch as remover pollutant in different ways.

**Figure 6 polymers-13-01737-f006:**
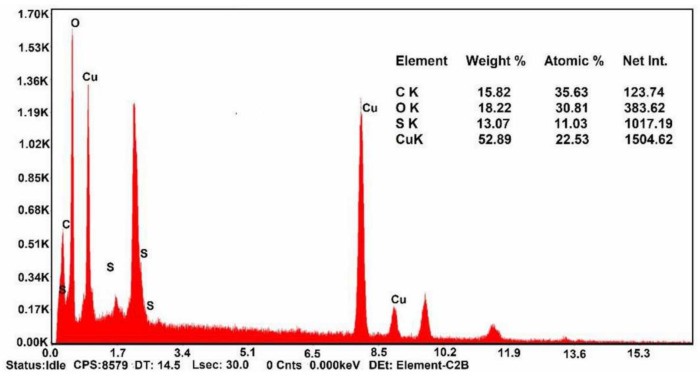
Energy-dispersive X-ray spectroscopy (EDX) elemental analysis of cellulose nanofiber (CNF)-treated hydrogel after ion adsorption (initial Cu^2+^ concentration of 0.4 g/L, adsorbent dosage of 0.3 g/L, pH level of 5). (Reprinted from ref. [15]. Copyright 2019 with permission from Elsevier).

**Figure 7 polymers-13-01737-f007:**
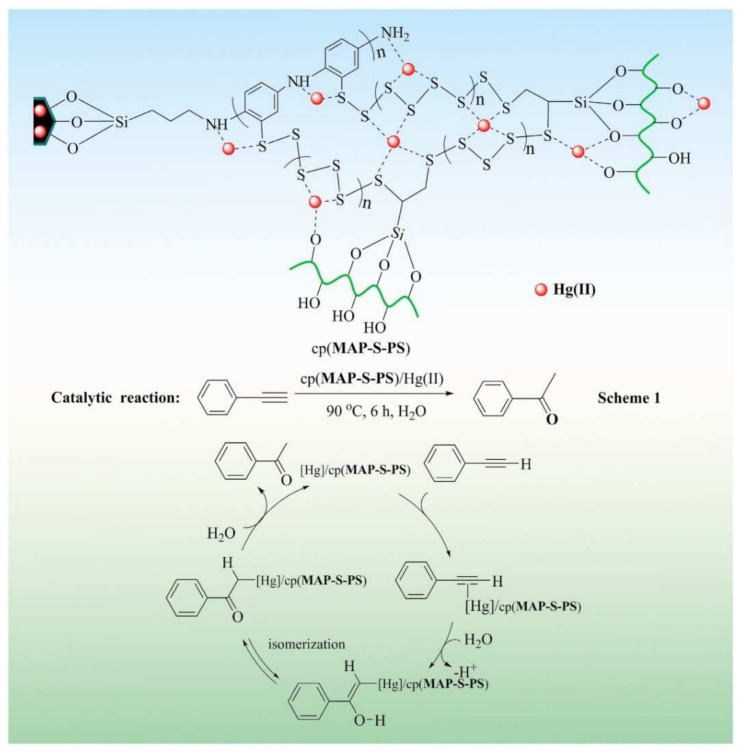
The adsorption mechanism of Hg(II) ions removed by polysulfide complex material starch based. (Reprinted from ref. [134]. Copyright 2021 with permission from Elsevier).

**Figure 8 polymers-13-01737-f008:**
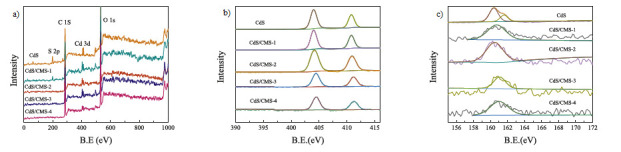
(**a**) Typical X-ray photoelectron spectroscopy (XPS) wide scan spectra, (**b**) Cd 3d and (**c**) S 2p XPS spectra of CdS, CMS, CdS/CMS-1, CdS/CMS-2, CdS/CMS-3, and CdS/CMS-4. (Reprinted from ref. [135]. Copyright 2020 with permission from Elsevier).

**Figure 9 polymers-13-01737-f009:**
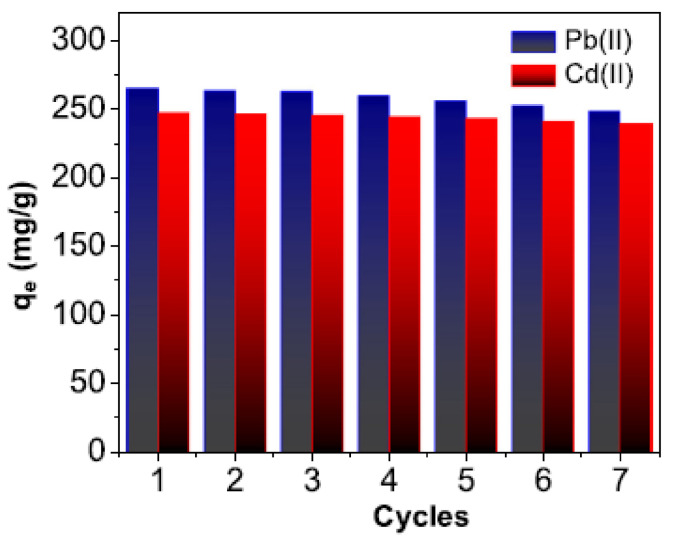
Reusability cycles of for the adsorption of Pb(II) and Cd(II) over starch-salicylaldehyde nanocomposites (2 mg, in 25 mL volume, time 60 min and the initial concentration 50 ppm). (Reprinted from ref. [122]. Copyright 2020 with permission from Elsevier).

**Figure 10 polymers-13-01737-f010:**
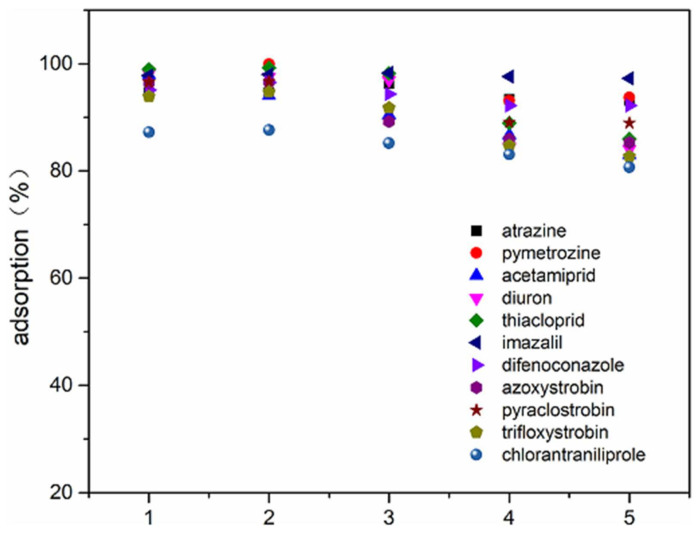
Adsorption efficiency of 11 pesticides by adsorbent mesoporous activated carbon from starch after various cycles of regeneration. (Reprinted from ref. [152]. Copyright 2019 with permission from Elsevier).

## Data Availability

Not applicable.

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
