# Peer review of "Two Fascinating Polysaccharides: Chitosan and Starch. Some Prominent Characterizations for Applying as Eco-Friendly Food Packaging and Pollutant Remover in Aqueous Medium. Progress in Recent Years: A Review"

_polymers, 2021, doi:10.3390/polym13111737_

Round 1

Reviewer 1 Report

The review is very well written. It is well organized and readers friendly. The authors have used updated references. Two observations are listed below

1: The paragraphs are too long. The authors should try to maintain a length of 10-20 sentences for a single paragraph.

2. The references used are so updated . mostly from 2019 to 2021 with few from the back years. Is it so that there is not much literature available on the subject in previous years or decades or the authors just want to stick to updated work. In the later case i will suggest to add a time frame to the title. for example the author may wish to cite developments on the subject from 2017 till date. So this point should be added to the title. A proposed title can be  : Two fascinating polysaccharides: chitosan and starch. Some prominent characterizations done in recent years (e.g 2017-2021 whatever time frame the  author chose) for applying as eco-friendly food packaging and pollutant remover in aqueous medium: A review 

Author Response

Dear reviewer 1,

I attach the responses to your valuable comments.

Reviewer 2 Report

This paper reports on the “Two fascinating polysaccharides: chitosan and starch. Some prominent characterizations for applying as eco-friendly food packaging and pollutant remover in aqueous medium: A review”. The article is interesting. Manuscript seems be corrected.

I have few comments to the manuscript:

  1. Missing reference in first paragraph.
  2. In the subsection with XPS, DSC and TGA, it would look good to add example charts.
  3. Figure 6 It should have been in an earlier chapter.
  4. Page 14 line 637. Corrected “Fe3O4”
  5. Page 14 line 668. Deleted extra space.
  6. There are no references in places where the authors use their own research and these are not summaries of previously described paragraphs with references e. g. Page 15 line 711-724
  7. Add a chapter about “Starch” basic information.

Taking into account all comments the manuscript may be published in Polymers after minor revision.

Author Response

Dear reviewer 2:

I attach the responses to your valuable comments.
